# Dealing with being prescribed cardiovascular preventive medication: a narrative analysis of qualitative interviews with patients with recent acute coronary heart disease in Sweden

Josabeth Hultberg  ,[1,2] Staffan Nilsson,[2] Carl Edvard Rudebeck,[3] Anita Kärner Köhler[2]

¹Åby Primary Health Care Centre, SE-616 21 Åby, Sweden
²Department of Health, Medicine and Caring Sciences, Linköping University, SE-581 83 Linköping, Sweden
³Research Unit, Kalmar Region, Kalmar, Sweden

**Correspondence to**
Dr Josabeth Hultberg;
josabeth.hultberg@gmail.com

## ABSTRACT

**Objective** To explore how patients with experience of acute coronary heart disease make sense of, and deal with, the fact of being prescribed cardiovascular preventive medication.

**Design** Qualitative interview study.

**Setting** Swedish primary care.

**Participants** Twenty-one participants with experience of being prescribed cardiovascular preventive medication, recruited from a randomised controlled study of problem-based learning for self-care for coronary heart disease.

**Methods** The participants were interviewed individually 6–12 months after their hospitalisation for acute coronary disease. A narrative analysis was conducted of their accounts of being prescribed cardiovascular preventive medication.

**Results** Four themes shape the patients' experiences: *'A matter of living'* concerns an awareness of the will to live linked to being prescribed cardiovascular preventive medication regarded in the light of the recent hospitalisation. In *'Reconciliation of conflicting self-images'*, patients dealt with being prescribed preventive medication through work to restore an identity of someone responsible in spite of viewing the taking of medication as questionable. The status of feeling healthy, while being someone in need of medication, also constituted conflicting self-images. Following this, taking medication was framed as necessary, not as an active choice. *'Being in the hands of expertise'* is about the seeking of an answer from a reliable prescriber to the question: 'Is this medication really necessary for me?' Existential labour was done to establish that the practice of taking cardiovascular preventive medication was an inevitable necessity, rather than an active choice. *'Taking medicines no longer a big deal'* could be the resulting experience of this process.

**Conclusions** Unmet existential needs when being prescribed cardiovascular preventive medication seem to be a component of the burden of treatment. A continuous and trustful relationship with the prescribing doctor may facilitate the reconciliation of conflicting self-images, and support patients in their efforts to incorporate their medicines taking into daily life.

## Strengths and limitations of this study

► Through narrative analysis of interviews, this study explores the patient perspective of being prescribed cardiovascular preventive medication, and how it is dealt with in everyday life.

► A wide range of experiences of being prescribed cardiovascular preventive medication is represented in the data following the selection of participants from a study of self-care after hospital care for acute coronary heart disease.

► Swedish healthcare has a lower proportion of general practitioners than similar countries, and only a minority of the population is registered with a regular general practitioner (GP), which may limit the transferability of the findings.

► The interviews were conducted within a year of an episode of acute coronary heart disease, so their accounts may be coloured by psychological reactions to the relatively recent serious event.

## INTRODUCTION

Cardiovascular preventive medication (CPM) is widely used and prescribed mainly in primary care. Patients tend to be reluctant to use medicines.[1 2] Patients with long-term medication manage their treatments every day. There is increasing consideration of the *burden of treatment*, the continuous and sometimes arduous work taken on by patients with long-term medication.[3 4] How patients formulate the pros and cons of CPM seems similar for primary and secondary prevention.[5] A framework has been proposed in which the perceived necessity to use medication is weighed against potential concerns.[2 6] Patients who are prescribed long-term medication navigate among paradoxical views on medication.[7] Seeing preventive medication as responsible and healthy behaviour stands

in contrast to the view that taking medicine may be dangerous, and a morally dubious alternative to healthy habits.[8 9] Patients' arguments against taking CPM include that they rely on their body's natural processes, and do not want to disturb it with medication.[10] Patients who use CPM in spite of their scepticism towards it argue that they are 'pill-takers' out of necessity.[9] Being prescribed long-term medication, also for prevention, may lead to a changed self-perception that invokes a rewriting of one's biography.[11 12] Patients display resistance, not only to *taking medication*, but also to *becoming someone who does*.[9 11 13] The resistance to taking medicines may be intertwined with a resistance towards adopting an identity as a chronically ill person.[8]

Patients' reluctance to take medicines, and their experienced efforts while doing so, must be taken into account by prescribers. Studies of communication in actual consultations shed light on how patients execute their agency in decision-making through subtle resistance to treatment proposals, but also how they appreciate the support of prescribers to express their preferences.[13 14] The inherent uncertainty of CPM prescribing is particularly challenging, as the treatment aims to reduce the abstract risk of future disease, not to relieve symptoms.[15 16] There is no unified definition of what shared decision-making (SDM) actually is, but its aim—to balance medical recommendations and the individual preferences of patients—makes great sense.[17] However, the assumption that a treatment decision is a distinct choice between neutrally formulated alternatives, occurring within the framework of a single consultation, has been contested.[18] Models that focus solely on the narrow context of individual consultations disregard the wider context of the everyday realities experienced by patients. This may cause *contextual errors* in decision-making,[19] which may increase the burden of treatment.[3] Common interpretations of SDM involve simplistic applications of autonomy ethics, and neglect the inherent asymmetry in patient–physician relationships.[20] It is not always realistic to expect patients to make independent rational decisions about medical treatments.[21] Decision aids, such as Systematic COronary Risk Evaluation[22] (SCORE) which explicitly quantify risk and include inherent uncertainties, have been notoriously difficult to implement in clinical practice.[23] Despite this, current guidelines recommend that SDM is used, and it has even been suggested that it should be mandatory.[22 24]

Physicians need guidance by relevant research to improve their skills in prescribing, which include patient–doctor communication, and decision-making about treatments in the consultation.[25] Studies of prescribing in consultations are necessary but not sufficient.[13 14 18] Understanding how prescriptions of CPM play out in patients' lives will help physicians fulfil their role as prescribers.[3 19] This study aimed to explore how patients with experience of acute coronary heart disease make sense of, and deal with, the fact of being prescribed CPM.

**Table 1** Participant characteristics, N=21

| Characteristics | n |
|---|---|
| **Sex** | |
| Male | 17 |
| Female | 4 |
| Age, years | 72 (52–86)* |
| **Residential area** | |
| City | 8 |
| Rural or small town | 13 |
| Education | |
| Compulsory education† | 8 |
| Upper secondary school | 6 |
| University | 7 |
| Marital status | |
| Cohabitating | 16 |
| Living alone | 5 |

*Median (min–max).
†Less than 10 years in school.

## METHODS
### Setting and participants
Participants were recruited consecutively from COR-PRIM, a primary care study of self-care after acute coronary heart disease.[26] In this population, there are experiences of previous prescriptions of CPM both for primary and secondary prevention, making it suitable for our study aim. Out of 29 approached by telephone, 22 accepted to be interviewed. One interview was excluded because the audio-recording failed. See table 1 for participant characteristics.

The sample size was preliminary set a priori to around 20, based on the method used.[27] In analysing, we found the data set do be rich and varied, and decided not to include more participants. All participants had been admitted to hospital for acute coronary heart disease 6–12 months prior to the interview, and gave written consent to participate after receiving oral and written information about the study. The wives of six of the participants were present during the interviews, and supported their husbands to recall events and express their experiences. The participants were interviewed in their homes by JH. She is a general practitioner with no prior relation to any of the participants. They were asked to recall a situation in which cardiovascular preventive drugs had been discussed or prescribed, without distinguishing between primary and secondary prevention. Drugs considered were antihypertensive medication, statins and antiplatelet drugs such as acetylsalicylic acid. Follow-up questions were asked to guide towards relevant topics and to explore them in greater depth. Figure 1 presents the interview guide. The interviews lasted for 45–90 min, and were audio-recorded and transcribed verbatim by JH.

Topics to cover in interviews:

- Cardiovascular risk, risk factors, the concept of risk and the participant's own risk of cardiovascular disease

- Power, responsibility and participation in prescribing decisions

- Knowledge relevant for treatment decisions

- Communication between patients and healthcare professionals about medication

**Figure 1** Interview guide.

## Analysis

We chose a narrative approach to analyse patients' accounts of being prescribed CPM.[27 28] The analysis concerned both content (*what* was told), and structure (*how* it was told) as described by Reissman.[27] In the initial reading of the transcriptions, we looked for salient features of the material as a whole, with a narrative framework in mind. Structural elements (figure 2) served to identify narratives.[28] The prescription of CPM was the complicating action around which these narratives revolved. The narratives described both the prescription of CPM after a recent cardiovascular event, and previous occasions on which CPM had been prescribed. Some interviews contained more than one narrative. A synopsis of each interview was made, in which the narratives were reproduced in condensed form. These condensed narratives were analysed in greater depth to define experiences, deliberations and actions of the participants related to the fact that they were being prescribed CPM. Structural features such as the framings and evaluations of the narrative material displayed their sense-making of reported speech.[28]

The narratives were rich in quotes. The participants told about what had been said, but also what may be said in a future scenario, by characters such as prescribers, other physicians, nurses, relatives and friends. They quoted texts, such as information leaflets. They also quoted themselves, both actual utterances, and their thoughts: their inner voices. Opinions and experiences of phenomena in the narratives became evident through internal evaluations.[28] Some, for example, used a mocking tone of voice

| Abstract | introduction of story |
|---|---|
| Orientation | background information, setting and participants in story |
| Complicating action | account for events in temporally ordered narrative clauses |
| Evaluation | revelations of attitudes towards the content of the story, what the point of the story is according to the narrator, why it makes sense, how it creates meaning |
| Resolution | the end result of the events |
| Coda | concluding remark |

**Figure 2** Structural elements in narratives according to Labov and Waletsky.

when quoting someone they did not agree with. The analysis of the different voices used by the participants rested on Goffman's analytical framework for aspects of narrators, including the concept of 'footing'. This framework helps to reveal the speakers' positioning of themselves in relation to reported speech and events.[29]

Tentative themes within one interview were assessed against each other, and then against those from other interviews. Similar tentative themes were brought together into more general themes, which were then further elaborated in an iterative process. Finally, the themes were checked for their bearing on the original interview data, and adjusted to provide the resulting themes. In each step, JH made an initial analysis. All authors discussed the emerging results. Throughout the process, the analytical focus was to capture the participants' ways of dealing with the fact that they were being prescribed CPM.

### Patient and public involvement

No patient involvement.

## RESULTS

The interviews were numbered #1–21, in the order they were conducted. The participants' medical histories ranged from no CPM prescriptions prior to the recent coronary event (#1), to over 20 years since the first myocardial infarction (#7), and many years with primary preventive medication with (#15) or without type two diabetes (#11). Four themes became apparent during the analysis: *'A matter of living'*, *'Reconciliation of conflicting self images'*, *'Being in the hands of expertise'* and *'Taking medicines no longer a big deal'*. Quotes, including longer passages, were selected to illustrate the themes, and to display the dialogic and narrative structure of the data.

### A matter of living

Implicit in the recommendation for preventive treatment is its importance for survival. Such a message may evoke an awareness of being mortal, and the fundamental will to live. Dealing with the awareness of mortality is not merely about avoiding death, but involves reflections about what makes life worth living, and what it means to lead a meaningful life. Staying alive to continue relationships with loved ones was salient:

I didn't use to take that much medication but now I have to.

Interviewer: Who makes you?

The COPD (chronic obstructive pulmonary disease) and heart

Wife: Basically, he wants to live …

… with you! (looks at wife) (#12)

The awareness of being mortal that was evoked by being prescribed CPM stimulated efforts to strike a balance between the pros and cons of taking medication.

For example, side effects could be accepted with the argument that the medication was regarded lifesaving:

> It is important that I take this and I don't care if it makes me feel a bit sick, I have to or else I will die. (#14)

Some participants held fundamental objections to taking CPM, but even so experienced gratitude for having the chance to stay alive, which was attributed to the medication:

> … Now you're a bit scared and think: well, I'd better take it for the heart, that's what they said … so of course you're glad for all these pills and that they work in the body. (#2)

### Reconciliation of conflicting self-images

Most participants were generally averse to taking medicines, and held the view that they are generally overprescribed and overused. The participants typically presented themselves as sceptical towards medication, not prone to using it, and stated that medicines should only be taken if necessary. Being given a prescription and becoming a pill-taker challenged this self-image. Most participants had been prescribed CPM for primary or secondary prevention, prior to the recent hospitalisation for acute coronary heart disease. Both participants who had been prescribed medication for primary prevention before having had symptomatic cardiovascular disease, and those who did not have it until after such disease, became aware that they were at risk of future cardiovascular disease. They perceived that their status changed from healthy to being in need of correction through medical treatment:

> (Now) I have to. I take it for self-preservation, but like I say: I don't like to put chemicals into my body. (#3)

One way of dealing with the new conditions was to seek to preserve the 'only-when-necessary' view of self. Taking CPM was framed as an inevitable consequence of the state of health, not as an active choice. A face-saving rhetoric indicated that this was a sensitive topic, which evoked feelings of embarrassment and shame. Potential blame for having caused the state of being at risk was countered by a defensive line of argument. Participants emphasised that they had been struck by disease, or had high cholesterol values, in spite of a healthy lifestyle. Having smoked was mentioned, but downplayed: 'I used to smoke but I quit … I used to drink a lot of milk … I think that was the biggest cause' *(#21)*. Hard work, and stressful life events were common explanations. Heredity was another: 'These bloody sickness-genes' *(#11)*. One line of defence was to claim to be an exception in comparisons with others: 'Many get way too much medicine, but that's perhaps more in nursing homes' *(#4),* while another was to normalise the condition:

> Now it (angioplasty) is common, they go: 'Oh you did it too?' … In the nineties you went down in the

basement of the University Hospital. It was like coming to Hades in the underworld and was serious as hell. They did a couple a week, now they do several a day. *(#7)*

### Being in the hands of expertise

A personal relationship with a prescriber was essential for patients in their work to seek reassurance that CPM was necessary. Some had experienced a continuous relationship with the physician responsible for prescribing their medication, but this was not the case for most. Without such a relationship, patients experienced feelings of abandonment:

> They took away furosemide and that other one, and I lacked … they just said: 'You can see how you feel', but how and when, with whom should I talk about how I feel? I didn't have anyone to talk with; it was all up to me. *(#15)*

Being in the hand of expertise and *feel* powerless may not equate to *being* powerless, when patients take a position in which they actively accept being cared for. Patients may choose to place the responsibility for decision-making on the prescribing doctor if they have a trustful relationship with the doctor, and in this way resolve their doubts about whether it is necessary to take medication:

> You look up to someone who knows everything about this, and you are grateful that someone sort of weighs and measures you, and corrects and changes the medicines. It's like going to the dentist. You're helpless. What if the dentist says: 'Hey this tooth has to come out'. You don't ask: 'Do I really have to?'. Or at the garage, you don't have a clue; the car mechanic can say anything. So it feels a bit like you are powerless. *(#1)*

The question 'Do I really have to take it?' occurred frequently in the narratives. Reassurance was sought in dialogue with a prescribing physician, who could explain sufficiently clearly why the medication was recommended, and who was willing to take the responsibility to state that it was necessary. The attitude towards the authority of expertise ranged from more-or-less considered acceptance and trust: 'You have to trust what they say, and they know that a lot better than I do anyway.' *(#4)*, to critical scepticism: 'It is in my nature. I don't just accept anything' *(#16)*. Many narratives illustrated how important it was to have an expert to turn to, regardless of how the patient viewed the authority of expertise. This was particularly the case when such an expert was lacking. Some patients were aware of treatment guidelines, and that the expert community has different views on treatment indications. In spite of this, and in spite of an expressed request for information, patients described themselves to be in the hands of the medical expertise. Most were content with simply the possibility of dialogue: 'You don't pick a fight about it and say: 'I won't take this', or: 'I want this that

I read about'. No, you take it for granted that this is a competent person.' *(#19)*

## Taking medicines no longer a big deal

The concluding remarks in many narratives showed that an attitude in which taking medicines was no longer a big deal had been acquired. Medication had been incorporated into ordinary life:

> They said you had to take this … for everything to work and for the blood clots that can build up and all that, and I have brochures about it that I can read if I want, but I think now when you have them sitting there (nods towards the medicine packages on the kitchen shelf) it goes automatically, every night and every morning. (#2)

For some, a no-big-deal attitude came with minimal effort or reflection, whereas most had put considerable work into dealing with being prescribed CPM. This process of becoming a pill-taker without it being a big deal seemed to be facilitated by the resolution of existential challenges, the reconciliation of conflicting self-images, and the conclusion, after dialogue with a trusted prescriber, that it was necessary. The recent coronary heart event further lessened any hesitation about taking CPM, and motivated the view that it was truly necessary:

> Nobody had a clue before this happened or talked about me belonging to a risk group, but *now* after getting the stent, I obviously do, so that is something to think about (when deciding about cholesterol treatment). (#19)

Some patients who were still in the process of dealing with one or several aspects of being prescribed CPM experienced their medication as a reminder of their vulnerability. It was difficult to ignore these feelings. Taking the medicines was still burdensome, and not yet an effortless daily routine:

> (Shows that she has put the medicine dispenser out of sight in a cupboard) Sometimes when it is time for medication, and there are people over, I think I mustn't forget to take it when they have left, because I would never … Yak, it's so dreadful … Perhaps it takes a year to grasp, before everything has calmed down, all of this, and you make it sort of into a ritual after a while. (#1)

## DISCUSSION

Dealing with being prescribed CPM involved the themes: *A matter of living, Reconciliation of conflicting self-images, Being in the hands of expertise* and *Taking medicines no longer a big deal*. Patients oriented towards living rather than survival, and to find a coherent view of self as someone who must take CPM. The decision about CPM was framed as a question about whether it was necessary, not as a patient choice. Patients sought an answer from a reliable prescriber to the implicit question: Is this medication really necessary for me? This implies that patients may execute their autonomy by actively placing the responsibility for the decision about CPM with the prescriber. A successful process resolved doubts about if taking CPM was necessary and seemed to facilitate the incorporation of it into everyday life. A stalled process of making it necessary appeared as a particular burden of treatment.

The wide range of medical histories represented is a strength of this study. Previous prescriptions, both for primary and secondary prevention, were contrasted to the most recent, and reflected in the light of the recent hospitalisation, which contributes to the credibility of the results. The participants probably still experienced reactions to the serious event, which may have resulted in their focus on existential matters.[30] Apart from the connection between the recent hospitalisation and secondary preventive medication the patient experiences from primary and secondary prevention were similar. This is in line with previous studies of patient perspectives on CPM.[5 6 9] The COR-PRIM participants rated high on scales that measure patient empowerment and general self-efficacy.[31] They may be inclined to share their viewpoints to improve healthcare, and particularly well-suited to give voice to general experiences of receiving prescriptions for CPM, although possibly not representative of less empowered patients. The interviews were rich in content, with reflections on experiences of being prescribed CPM that may be transferable to similar contexts. Non-adherence was disclosed, and negative experiences of encounters with doctors reported, indicating that the participants felt trust, and did not adjust their stories to please the interviewer. All themes appeared in almost every interview, suggesting that the themes have a general relevance. The presence of partners in six interviews was not planned, but turned out to be advantageous. As noted in a similar study, the dialogues of the couples added perspectives to the accounts, while keeping a respectful focus on the patient's experiences.[9] Some participants showed signs of cerebrovascular disease with slight cognitive impact. This is to be expected in this group of patients. It is important, and a strength of the study that also their voices are heard. The presence of partners facilitated this. We believe that one reason to participate in COR-PRIM was the lack of a personal relationship with a prescribing physician that many participants described. Healthcare in Sweden has a lower proportion of general practitioners than similar countries, and only a minority of the population are registered with a regular GP.[32] This may reduce the transferability of the findings. On the other hand, the negative experiences of lacking a continuous personal relationship with a prescriber that the participants voiced suggest that this is of general importance.

The general reluctance towards medication was overridden by the will to live, with an understanding of taking CPM as necessary to survive. It was framed as an inevitable must, not as a choice. Polak described this standpoint as a defence of morality, that is, against the accusations of

leading an unhealthy life style.[9] This agrees with the defensive rhetoric that many of the participants took, and the way in which they dealt with the need for CPM as a morally sensitive issue. The framing of it being necessary to take CPM appeared also to overrule uncertainty, making it into a question of life or death, not quite consistent with the notion of choice between alternatives in commonly advocated SDM models. The effects of CPM are overestimated by both laymen and general practitioners, which may be a result of this concept of necessity.[6 16] Patients must deal with a challenge to the self-image, and self-perception changes when long-term medication such as CPM is prescribed.[11] Some of the resistance to taking medication may stem from a denial of being ill.[1] However, also medication prescribed for prevention, in the absence of symptomatic disease, may call for a rewriting of the biography.[12] The key to the reconciliation of the conflicting images held by the patients in our study was the work with identity by which they established that it was necessary to take CPM, and that they had become a medicine-taker.

Our findings indicate that there are patients who want explanations about CPM, but want the prescriber to determine whether it is necessary. It agrees with previous work where patients prefer to take part in the decision-making process, but not to make the final decision.[33 34] Patients can execute their agency and autonomy by asking for information and a dialogue but deferring the responsibility for the decision.[13] Prescribers must be aware of the inherent asymmetry in a patient–doctor relationship, and be receptive to the patients' active deferral of responsibility for decision-making. They must strike a balance: to avoid violating patient autonomy without abandoning them.[35 36] Patient autonomy in this situation is not absolute, but depends on the response of the prescriber.[20 37] Elderly patients who take several medicines are more likely to find the medication necessary if they interact with a 'good' doctor.[33] Even healthcare professionals, when patients, depend on the relationship with doctors and nurses, find rest in their hands, and trust that they know what they're doing.[38] Plain information from a faceless expert community causes even better-informed patients to feel abandoned.[39] The process of making the taking of CPM to be necessary may result in the taking being incorporated into ordinary everyday life, even though it may threaten the 'ordinariness' for persons who view themselves as non-pill-takers, and those who view medicines as a necessary evil.[8 9]

The implicit or explicit question: 'Is it really necessary for me to take CPM?' requires a definitive answer. The submission to expertise may be a conscious and autonomous act by which responsibility for decisions is placed on the professional. A personal continuous relationship with the prescriber helps patients to express their preferences and needs during the process that leads to a decision, and eventually to a no-big-deal attitude towards taking CPM. As Julian Tudor Hart put it: 'Effective health care has to be built around real continuing personal stories, not episodic fragments of standardized process'.[40] In the same spirit, the concept of shared understanding has been proposed, in an attempt to merge SDM, evidence-based medicine and patient-centred medicine. Attentive listening to patients' narratives is essential to the practice of shared understanding.[25 41]

## Conclusions

Based on the themes *'A matter of living'* and *'Reconciliation of conflicting self-images',* we suggest that unmet existential needs when being prescribed CPM is a component of the burden of treatment. Inferred from the themes *'Being in the hands of expertise'* and *'Taking medicines no longer a big deal',* we propose that a continuous and trustful relationship with the prescribing doctor may facilitate the reconciliation of conflicting self-images, and support patients in their efforts to incorporate their medicines taking into daily life. These findings should be investigated in larger populations for increased knowledge of how common the different ways of dealing with being prescribed CPM are in different groups of patients.

**Acknowledgements** We are indebted to the participants for their generosity with their time and stories, and for their hospitality during the interviews.

**Contributors** All authors were responsible for the planning and design of the study. AKK and SN were responsible for the recruitment of participants. JH performed and transcribed the interviews. All authors contributed to the analysis. JH wrote a manuscript draft, which CER, AKK and SN critically revised. JH is the corresponding author and responsible for the over all content. All authors approve of the submission of this manuscript and agree to be accountable for all aspects of the work in ensuring that questions related to the accuracy or integrity of any part of the work are appropriately investigated and resolved.

**Funding** This study was supported by the County Council of Östergötland (no grant number) and Lions forskningsfond mot folksjukdomar (Lions Sweden, Districts 101-V, 101-S, 101-O and Linköping University Medical Faculty, no grant number).

**Competing interests** None declared.

**Patient and public involvement** Patients and/or the public were not involved in the design, or conduct, or reporting, or dissemination plans of this research.

**Patient consent for publication** Consent obtained directly from patient(s)

**Ethics approval** This study involves human participants and was approved by Regional Ethics Committee of Linköping, Ref. No. 2014/5-31. Participants gave informed consent to participate in the study before taking part.

**Provenance and peer review** Not commissioned; externally peer reviewed.

**Data availability statement** Data are available upon reasonable request.

**ORCID iD**
Josabeth Hultberg http://orcid.org/0000-0002-5107-6131

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
