## [Reviewer comments · BMJ Open]

ARTICLE DETAILS

TITLE (PROVISIONAL)	Dealing with being prescribed cardiovascular preventive medication: a narrative analysis of qualitative interviews with patients with recent acute coronary heart disease in Sweden.
AUTHORS	Hultberg, Josabeth; Nilsson, Staffan; Rudebeck, Carl Edvard; Köhler, Anita Kärner

VERSION 1 – REVIEW

REVIEWER	Nishigaki, Nobuhiro Takeda Pharmaceutical Company Limited, Japan Medical Office
REVIEW RETURNED	03-Oct-2021

GENERAL COMMENTS	The authors have undertaken a carefully designed qualitative study to explore what patients with experience of acute coronary heart disease consider and want when they are prescribed cardiovascular preventive medication. They identified four themes. This study is novel and has a potential to provide valuable insights into the perception of patients who are prescribed cardiovascular preventive drugs and what is important for prescribers to establish treatment alliances with the patients. Despite these advantages, it is my opinion that this article requires significant revisions to ensure acceptance for publication. I outline my specific concerns below: 1. Abstract Conclusions include inferences that are not based on results, so conclusive expressions should not be used. It should also be noted that further investigation is needed to draw appropriate conclusions. 2. Method section The objective of this study was “to explore how patients with experience of acute coronary heart disease make sense of, and deal with, the fact of being prescribed cardiovascular preventive medication.” The participants were prescribed cardiovascular preventive medication. In the Materials and Methods section, it is not appropriate to state "no patient involvement". I presume that participants may not have had coronary heart disease, but had lifestyle diseases and comorbidities. 3. Result section •Many confounders exist when exploring patients' perceptions on being prescribed cardiovascular preventive drugs. The more detailed participant characteristics should be provided in this report as COR-PRIM study described them (i.e. diseases which participants have, smoking status, job position, Canadian
---

	Cardiovascular Society scale, types of cardiovascular preventive medications and duration of preventive drug prescription).  •It would be easier for readers to understand which themes are important if you could show what percentage of participants have experienced each of the four themes. It is also important to show that the authors did not select only convenient narratives.. •It would be better to examine and describe the relationship between the experience of each four themes and whether or not you have already encountered a reliable prescriber, whether or not you felt anxiety or dissatisfaction during shared decision making, or whether or not you obtained an appropriate response to the question 'Is this medication really necessary for me?' from the prescriber. 4. Discussion section Limitations of this study should be discussed. A limited number of patients participated. I think there is a selection bias of participants. The participants were elderly, few have high education, and many live in rural areas. In addition, participants were recruited consecutively from COR-PRIM study, a randomized controlled study of problem-based learning for self-care for coronary heart disease, which more than half of eligible patients declined to participate. Recall bias may also exist if participants were older and included patients with cerebrovascular disease.
--	---

REVIEWER	McKinn, Shannon The University of Sydney, Sydney School of Public Health
REVIEW RETURNED	28-Oct-2021

GENERAL COMMENTS	Thank you for the opportunity to review this interesting paper. General: 1. When reading the Introduction, I was a bit confused about whether you were talking about primary or secondary prevention of CVD, as some of your points and references in this section are quite obviously about primary prevention, while the manuscript is quite obviously about secondary prevention, given the title. I understand that you had reasons for not making this distinction in your analysis, as mentioned in the Discussion. However, the lack of delineation between primary and secondary when setting up the background and rationale for this study is a bit confusing and potentially undermines your argument for SDM, as this comes in directly in response to a sentence about the inherent uncertainty of CPM prescribing based on a risk calculation rather than a symptomatic condition. I think it would be useful to the reader for the authors to make this distinction in the introduction, and to include your rationale for not making this distinction in your analysis (because patients tend not to make this distinction) earlier in the manuscript. You may also wish to include this reference in support of your decision to not differentiate between patients' experiences of taking medications for primary and secondary prevention, as it also found participants experiences to very similar across both (10.1136/bmjopen-2018-026342) Abstract: 2. I was a bit confused reading the sentence describing your second theme "Reconciliation of conflicting self-images". I think splitting this into two shorter sentences would help here. Intro:
--

	3. p3, line 7: Cardiovascular risk is not a chronic condition. Perhaps some factors that contribute to a person’s overall CVD risk could be classified as such, but risk is just risk, not a “condition.” Methods: 4. P4 line 41: “involved” should be “involvement” 5. Was the sample size decided a priori, or was there a consideration of data or thematic saturation? How was this determined? 6. Was JH known to participants through the larger study at all, was there any preexisting relationship? Were there any personal characteristics of the interviewer that may have impacted participant accounts? E.g. if JH is a doctor, participants might feel like that had to report that they are adherent with their medication. Results: 7. P7, line 30: word order “experienced even so” should be “even so experienced” 8. P7 line 44: suggest “using it”, not “use it” 9. P8, line 54: “Being in the hands of expertise and feel powerless may not be powerless,” – suggest “Being in the hand of expertise and feeling powerless may not equate to being powerless.” 10. I suggest that some of your longer quotes (i.e. those that are three lines or longer) should be started on a new line, rather than running straight on within the paragraph. Discussion: 11. P11, line 9: you mention the wide range of experiences displayed by participants as a strength of this study. This wide range is not evident in the results; rather it seems that their experiences, or at least their perceptions/sense-making of their experiences is quite similar across the sample. It’s also not clear if you are talking about their medical history or their experiences specific to being prescribed CPM. Perhaps an extra paragraph at the top of the results that describes this range of experiences/medical histories(?) more explicitly would be helpful. 12. “The participants in COR-PRIM rated high on scales that measure patient empowerment and general self-efficacy.” While you state that this may make participants particularly suited to giving voice to their experiences, how might this impact generalisability of your findings? 13. As SDM featured quite prominently in the introduction, I was surprised to see it almost entirely absent from your Discussion, particularly the second half of p12. This paragraph seems to be a critique of SDM and the current guidelines for using SDM which you mentioned in the introduction, without mentioning SDM. If that is the case, you should make this explicit.
--	---

VERSION 1 – AUTHOR RESPONSE

Reviewer: 1 comments and responses

Dr. Nobuhiro Nishigaki, Takeda Pharmaceutical Company Limited Comments to the Author:

The authors have undertaken a carefully designed qualitative study to explore what patients with experience of acute coronary heart disease consider and want when they are prescribed cardiovascular preventive medication. They identified four themes. This study is novel and has a potential to provide valuable insights into the perception of patients who are prescribed cardiovascular preventive drugs and what is important for prescribers to establish treatment alliances with the

patients. Despite these advantages, it is my opinion that this article requires significant revisions to ensure acceptance for publication. I outline my specific concerns below:

1. Abstract

Conclusions include inferences that are not based on results, so conclusive expressions should not be used. It should also be noted that further investigation is needed to draw appropriate conclusions. # response: Thank you for this very relevant comment about the expressions in the conclusions. We realize that they may be read as absolute statements, which was not our intention. Please see the conclusions paragraph in the end of the main document, which has been revised with mitigated expressions concerning what can be inferred from the results. We have also added a statement about further studies. P 16 lines 2-10. The conclusions section in the abstract has been replaced with an abbreviated version of the paragraph with the mitigated statements. See p 3 lines 1-5. The important question of further investigations and future research is also addressed in the responses to points 3 and 4 below.

2. Method section

The objective of this study was "to explore how patients with experience of acute coronary heart disease make sense of, and deal with, the fact of being prescribed cardiovascular preventive medication." The participants were prescribed cardiovascular preventive medication. In the Materials and Methods section, it is not appropriate to state "no patient involvement". I presume that participants may not have had coronary heart disease, but had lifestyle diseases and comorbidities. [NOTE FROM THE EDITORS: the comment about "no patient involvement" can be rebutted, as this is a journal requirement and refers to involvement in the design and implementation of the study. Hopefully the above editorial request to move this statement to the end of the Methods section will help to avoid confusion]

response: The required heading "Patient and public involvement" has been moved to the end of the methods section as advised.

3. Result section

.Many confounders exist when exploring patients' perceptions on being prescribed cardiovascular preventive drugs. The more detailed participant characteristics should be provided in this report as COR-PRIM study described them (i.e. diseases which participants have, smoking status, job position, Canadian Cardiovascular Society scale, types of cardiovascular preventive medications and duration of preventive drug prescription).

It would be easier for readers to understand which themes are important if you could show what percentage of participants have experienced each of the four themes. It is also important to show that the authors did not select only convenient narratives..

.It would be better to examine and describe the relationship between the experience of each four themes and whether or not you have already encountered a reliable prescriber, whether or not you felt anxiety or dissatisfaction during shared decision making, or whether or not you obtained an appropriate response to the question 'Is this medication really necessary for me?' from the prescriber. #response: Regarding participant characteristics: An addition is made in the top of the results section (P7, lines 25-29) , to describe the range of medical histories of the participants. We have deliberately refrained from adding more variables than those given in table 1 (participant characteristics) for ethical reasons as more detailed information would make possible an identification of participants. In addition, as came out of the interviews, they had similar main traits, although the medical details of the informants differed. We do therefore not believe that further detailing would alter the reception of the results.

Regarding analysis and report of results: We have attempted to deal with the risk of bias in the analysis by the stepwise process and checking of preliminary results among the co-analysts. Details were added to clarify this, please see the end of the analysis section, P7, lines 15-18. Illustrative quotes from the original data were provided in accordance with conventions for the reporting of qualitative studies. The qualitative approach of this study does not render suitable data for a quantified distribution of the themes nor for calculations of associations between the resulting themes and the (highly relevant!) variables that you mention. Further studies however, as you mentioned in point 1, are necessary to try the findings in a wider population. With a quantitative approach it would be possible to calculate the occurrence of a particular experience or way of dealing with being prescribed CPM, for example in an adequately powered questionnaire study. As mentioned above, an addition about future studies was made in the conclusions section, according to your suggestion (P16, lines 2-10). In such a study a larger number of patients could be asked, like you suggest, for example whether or not they recall obtaining an appropriate response to the question "is this medication really necessary for me?". The question of necessity of medication reported in the present study however, was a recurrent topic abstracted from the verbatim statements of the participants, and not always formulated by patients with that exact wording. To clarify and emphasize this, additions of "implicit" to the discussions around the question "is this really necessary for me?" has been made. See P12, line 4 and P14 line 20.

4. Discussion section

Limitations of this study should be discussed. A limited number of patients participated. I think there is a selection bias of participants. The participants were elderly, few have high education, and many live in rural areas. In addition, participants were recruited consecutively from COR-PRIM study, a randomized controlled study of problem-based learning for self-care for coronary heart disease, which more than half of eligible patients declined to participate. Recall bias may also exist if participants were older and included patients with cerebrovascular disease.

#response: As you point out, there was a risk of skewed selection of participants, since they were recruited from another larger trial, where a substantial proportion of eligible participants declined participation. Most who did so did not have the time (because they were working) or the energy (because they were too old and frail). The choice to recruit participants from the COR-PRIM-study was to ensure participation of patients with a range of experiences of being prescribed cardiovascular preventive drugs, which we considered important for the research question of the present qualitative study. Both younger patients still working, and older frail patients were represented in the sample. Please see additions in the materials and methods section, subheading "participants and setting" about deliberations regarding the sample and relevance of data: P5 lines 18-20 and P6, lines 4-6. We have also added a section to the discussion about selection of participants with possible impact on transferability: p12 lines 22-29. We acknowledge the issue you mention with older patients with cerebrovascular disease and would like to point to the discussion in P12, line 33 and P13 lines 1-5.

Reviewer: 2 Comments and responses

Dr. Shannon McKinn, The University of Sydney, The University of Sydney Comments to the Author:
Thank you for the opportunity to review this interesting paper.

General:

1. When reading the Introduction, I was a bit confused about whether you were talking about primary or secondary prevention of CVD, as some of your points and references in this section are quite obviously about primary prevention, while the manuscript is quite obviously about secondary prevention, given the title. I understand that you had reasons for not making this distinction in your analysis, as mentioned in the Discussion. However, the lack of delineation between primary and secondary when setting up the background and rationale for this study is a bit confusing and potentially undermines your argument for SDM, as this comes in directly in response to a sentence

about the inherent uncertainty of CPM prescribing based on a risk calculation rather than a symptomatic condition. I think it would be useful to the reader for the authors to make this distinction in the introduction, and to include your rationale for not making this distinction in your analysis (because patients tend not to make this distinction) earlier in the manuscript. You may also wish to include this reference in support of your decision to not differentiate between patients' experiences of taking medications for primary and secondary prevention, as it also found participants experiences to very similar across both (10.1136/bmjopen-2018-026342)

response: Thank you for this very suitable reference and for helpful suggestions on how to avoid confusion about primary and secondary prevention. The introduction has been rewritten to clarify - and argue for - the decision not to differentiate between primary and secondary prevention in the analysis, and in response to your point 3. Please see P3 lines 29-34 and P4 lines 1 (suggested reference added here) and 21-22. There is also an addition of the rationale for the selection of participants with regard to the focus on both primary and secondary prevention, in the second paragraph of the methods section, P5, line 19, and a comment in the discussions section, P12 lines 15-19, where the suggested reference is also added.

Abstract:

2. I was a bit confused reading the sentence describing your second theme "Reconciliation of conflicting self-images". I think splitting this into two shorter sentences would help here.

response: Yes, that was unclear., thank you for the suggestion. The sentence is split into two and rephrased for clarity. P2, lines 21-25.

Intro:

3. p3, line 7: Cardiovascular risk is not a chronic condition. Perhaps some factors that contribute to a person's overall CVD risk could be classified as such, but risk is just risk, not a "condition."

response: Yes, we agree, having risk factors is not a chronic condition. Rephrased, please see P3, lines 29-30.

Methods:

4. P4 line 41: "involved" should be "involvement"

response: Changed as suggested (and moved to end of methods section as advised by the editor and in response to comment from reviewer 1) P7, lines 21-22.

5. Was the sample size decided a priori, or was there a consideration of data or thematic saturation? How was this determined?

response: Thank you for a relevant question. We added a comment on the sample, please see P 6 lines 4-6.

6. Was JH known to participants through the larger study at all, was there any preexisting relationship? Were there any personal characteristics of the interviewer that may have impacted participant accounts? E.g. if JH is a doctor, participants might feel like that had to report that they are adherent with their medication.

#response: JH is a medical doctor, with no prior relation of any kind to the participants. Information added in methods section, see P6, lines 11-12. Non-adherence was disclosed in several cases, and reports of negative experiences of encounters with prescribers were frequent, indicating that the participants did not adjust their stories to please the interviewer, in spite of her being a doctor. Please see comment added in the discussion section, P 12, lines 26-29

Results:

7. P7, line 30: word order "experienced even so" should be "even so experienced"

#response: Changed as suggested. In the marked copy this is on page 8 line 23.

8. P7 line 44: suggest "using it", not "use it"

#response: Changed as suggested. In the marked copy this is on page 8 line 31

9. P8, line 54: "Being in the hands of expertise and feel powerless may not be powerless," - suggest "Being in the hand of expertise and feeling powerless may not equate to being powerless."

#response: Changed as suggested. Page 10, line 5.

10. I suggest that some of your longer quotes (i.e. those that are three lines or longer) should be started on a new line, rather than running straight on within the paragraph.

#response: Changed as suggested. P9 lines 21-22, P9, line 32 and P10 line1, P10, lines 9-10.

Discussion:

11. P11, line 9: you mention the wide range of experiences displayed by participants as a strength of this study. This wide range is not evident in the results; rather it seems that their experiences, or at least their perceptions/sense-making of their experiences is quite similar across the sample. It's also not clear if you are talking about their medical history or their experiences specific to being prescribed CPM. Perhaps an extra paragraph at the top of the results that describes this range of experiences/medical histories(?) more explicitly would be helpful.

#response: Thank you for the suggestion. A paragraph that describes and exemplifies relevant characteristics of the participants' medical histories (a more suitable term than experiences, thanks!) has been added at the top of the results section, P7, lines 25-29.

12. "The participants in COR-PRIM rated high on scales that measure patient empowerment and general self-efficacy." While you state that this may make participants particularly suited to giving voice to their experiences, how might this impact generalisability of your findings?

#response: Please see addition on P 12 lines 22-23 to address the limitations in terms of its possible generalisability.

13. As SDM featured quite prominently in the introduction, I was surprised to see it almost entirely absent from your Discussion, particularly the second half of p12. This paragraph seems to be a critique of SDM and the current guidelines for using SDM which you mentioned in the introduction, without mentioning SDM. If that is the case, you should make this explicit.

#response: Yes, there was an imbalance with a strong focus on SDM in the introduction and less so in the discussion. The concerns about SDM are rephrased and made more explicit in the discussion of the results, please see P13 lines 18-20. To underpin this, the finding of patients' "necessity framing" regarding the decision about cardiovascular preventive medication has also been included in the summary of results in the top of the discussion section, P12 lines 2-3.

VERSION 2 – REVIEW

REVIEWER	McKinn, Shannon The University of Sydney, Sydney School of Public Health
REVIEW RETURNED	29-Nov-2021
GENERAL COMMENTS	Thank you for your careful consideration of my comments. I have no further comments.